# Mitochondrial Dysfunction in Metabolic Dysfunction Fatty Liver Disease (MAFLD)

**DOI:** 10.3390/ijms242417514

**Published:** 2023-12-15

**Authors:** Ying Zhao, Yanni Zhou, Dan Wang, Ziwei Huang, Xiong Xiao, Qing Zheng, Shengfu Li, Dan Long, Li Feng

**Affiliations:** 1Division of Liver Surgery, Department of General Surgery, West China Hospital, Sichuan University, Chengdu 610041, China; zhaoying_2021@163.com (Y.Z.); yannizhoul@hotmail.com (Y.Z.); doc_wangsurgeon@163.com (D.W.); 13778066872@163.com (Z.H.); nessen2004@163.com (X.X.); zhengqingscu@163.com (Q.Z.); lishengfu@126.com (S.L.); longdan0707@wchscu.cn (D.L.); 2Regeneration Medicine Research Center, West China Hospital, Sichuan University, Chengdu 610041, China; 3NHC Key Laboratory of Transplant Engineering and Immunology, West China Hospital Sichuan University, Chengdu 610041, China

**Keywords:** MAFLD, fatty acid metabolism, oxidative stress, mitochondrial quality control, liver–gut axis, mitochondrial antioxidant

## Abstract

Nonalcoholic fatty liver disease (NAFLD) has become an increasingly common disease in Western countries and has become the major cause of liver cirrhosis or hepatocellular carcinoma (HCC) in addition to viral hepatitis in recent decades. Furthermore, studies have shown that NAFLD is inextricably linked to the development of extrahepatic diseases. However, there is currently no effective treatment to cure NAFLD. In addition, in 2020, NAFLD was renamed metabolic dysfunction fatty liver disease (MAFLD) to show that its pathogenesis is closely related to metabolic disorders. Recent studies have reported that the development of MAFLD is inextricably associated with mitochondrial dysfunction in hepatocytes and hepatic stellate cells (HSCs). Simultaneously, mitochondrial stress caused by structural and functional disorders stimulates the occurrence and accumulation of fat and lipo-toxicity in hepatocytes and HSCs. In addition, the interaction between mitochondrial dysfunction and the liver–gut axis has also become a new point during the development of MAFLD. In this review, we summarize the effects of several potential treatment strategies for MAFLD, including antioxidants, reagents, and intestinal microorganisms and metabolites.

## 1. Introduction

Nonalcoholic fatty liver disease (NAFLD) is the accumulation of fatty degeneration and lipo-toxicity due to intracellular lipid overload in the liver without alcohol, which further leads to liver fibrosis and finally evolves into nonalcoholic steatohepatitis (NASH) and liver cirrhosis [1,2,3]. In the past decade, NAFLD has become the main cause of hepatocellular carcinoma (HCC) in addition to chronic hepatitis B (CHB) and chronic hepatitis C (CHC) and is a common chronic liver disease in Western countries [4,5]. According to statistics, the global incidence of NAFLD has reached 25%, and it is associated with a higher risk of disease under the influence of basic metabolic diseases such as obesity and type 2 diabetes (T2DM) [6,7,8]. In 2020, the International Expert Group issued a statement proposing to rename NAFLD to metabolic dysfunction fatty liver disease (MAFLD) and rename NASH to metabolic dysfunction-associated steatohepatitis (MASH) [9,10]. This means that its pathogenesis is closely related to metabolic disorders. However, there is currently no effective treatment that can reverse or cure MAFLD or MASH in the clinic. According to studies and reports, the pathogenesis of MAFLD is diverse and complex, and there is no complete and systematic conclusion about the pathogenesis of MAFLD.

Oxidative stress and insulin resistance are recognized as hallmarks of MAFLD [11]. At the same time, metabolic syndrome, which includes conditions such as mitochondrial dysfunction and oxidative stress, inherent immune regulation disorder, and abnormal regulation of autophagy, was found to be an important factor affecting the development of MAFLD in clinical patients [2,11,12,13,14,15,16,17,18,19,20]. For example, An, P. et al. found that the copy number of mitochondrial DNA (mt-DNA) was significantly increased in MAFLD patients [21]. Pirola, C.J. et al. and Einer, C. et al., through liver biopsy of MASH mice, revealed morphological changes such as volume expansion of hepatocellular mitochondria, rounding of cristae, enhanced fluidity of the mitochondrial membrane, and loss of typical dense mitochondrial granules. In addition, they also found that the development of MAFLD was regulated by the transcriptional activity and surface modification degree of mt-DNA [22,23]. In recent years, intestinal microorganisms have also received widespread attention [24]. Rao, Y. et al. found that intestinal microorganisms can cause lipid metabolism and electron transport chain damage by inhibiting the secretion of short-chain fatty acids (SCFAs), thereby affecting the progression of MAFLD in rodents [25].

Mitochondria are the main sites of intracellular energy generation and oxidative metabolism of carbohydrates and fatty acids in cells by affecting various physiological mechanisms in MAFLD. Therefore, we have listed serval differences in mitochondrial dysfunction in chronic liver diseases. This review focuses on the causes of mitochondrial dysfunction in MAFLD and the mechanism by which mitochondrial dysfunction affects the conversion of MAFLD to HCC. The main therapeutic strategies about mitochondrial functions for MAFLD and other chronic liver diseases are summarized in Table 1.

## 2. Mitochondrial Structure and Function in MAFLD

### 2.1. Mitochondrial Membrane Structure

The double-membrane structure of mitochondria provides a place for a variety of metabolic reactions. Porins distributed in the outer membrane are screening sites for metabolic substrates in the cytoplasm, and ion channels or binding proteins distributed in the inner membrane, including respiratory chain proteins, ATP synthases, and vitamin-binding receptors, are highly involved in mitochondrial metabolism. Recently, it has been found in clinical patients and mouse models of MAFLD that abnormal activation of a special type of nonspecific ion channel in the mitochondrial membrane structure, the mitochondrial permeability transition pore (MPTP), is associated with oxidative stress and the development of MAFLD [32,41,42,43].

In the MAFLD mouse model, it was found that intracellular accumulation of free fatty acids (FFAs) could directly stimulate MPTPs to maintain an open state, and mitochondrial Ca^2+^ outflow stimulated related inner membrane proteins, such as adenine nucleotide translocator or F1F0-ATPase, to aggregate, forming new MPTPs [32]. Moreover, this result was also found in MAFLD patients whose MPTPs could also be activated by mt-ROS and other oxidation byproducts [42,44].

Under mitochondrial stress conditions, MPTPs open in inner mitochondrial membranes. Increased membrane permeability leads to Ca^2+^ escape from mitochondrial calcium stores and electron transport chain (ETC) protons into the cytoplasm, resulting in typical MAFLD symptoms such as defective ATP production and increased cytoplasmic concentration of Ca^2+^ [45]. Moreover, significant mitochondrial swelling was found in MAFLD mice, including loss of the mitochondrial outer membrane, disappearance of cristae, and expansion of the mitochondrial matrix. These structural changes may also be caused by the abnormal activation of MPTPs through the action of FFAs [32]. MPTPs are widely involved in cell apoptosis and damage clearance mechanisms, which are related to an increase in membrane permeability leading to the release of cytochrome C into the cytoplasm. In addition, the activation of MPTP reduces the activity of mitochondrial inner membrane proteins and inhibits respiratory chain function and ATP synthesis, thus causing mitochondrial oxidative stress. Reduced oxidation activity of ETC-related complexes Ⅰ and III, cytochrome C, and coenzyme Q on the inner mitochondrial membrane leads to electron leakage, which then combines with oxygen molecules to form mt-ROS [46]. In MASH fibroblasts, mt-ROS expression and mitochondrial membrane permeability were found to be significantly increased and eventually escaped to other organelles [43].

A variety of functional proteins are located in the inner mitochondrial membrane. For example, abnormal activation of uncoupling proteins (UCPs) on the inner mitochondrial membrane could significantly reduce the mitochondrial membrane potential and H_2_O_2_ release, which are significant for the promotion of fat oxidation metabolism [30]. It is worth noting that decreased membrane potential led more free fatty acids to enter the mitochondria by decreasing the activity of inner proteins in oxidative phosphorylation (OXPHOS) and β-oxidation, resulting in lipotoxic accumulation to promote the deterioration of MAFLD toward MASH [47].

### 2.2. Mitochondrial DNA Mutation

In hepatocytes, more than 90% of mitochondrial proteins are encoded by nuclear DNA, while mitochondrial DNA (mt-DNA) encodes the remaining mitochondrial proteins. The expression of mt-DNA directly affects the efficiency of cellular oxidation. According to the genome comparison of clinical patients, the liver mitochondrial circular gene (mt-DNA) of MAFLD patients has a higher gene mutation rate and heterogeneity [31].

Mitochondrial cytochrome B, encoded by mt-DNA, is an essential component of complex III of the electron transport chain, which is responsible for electron transport and assisting in the formation of the proton gradient. In patients with MAFLD, it was found that the mt-CYB gene was mutated abnormally. The abnormal mt-CYB gene leads to ETC disorder by changing protein activity, stimulating the production and release of mt-ROS and carcinogenic metabolic byproducts (such as 2-hydroxyglutarate), and then aggravating the degree of MAFLD [48]. Existing studies have shown that increased mutations in mt-DNA and mt-DNA integrity may be associated with oxidative damage in vivo. ROS and peroxyl free radicals oxidize guanosine to a DNA oxidation complex (8-OHdG) and lipid peroxyl free radicals (4-HNE), respectively, to destroy the mt-DNA structure. Then, the damaged mt-DNA reduces ETC activity, which downregulates the OXPHOS response and participates in the pathogenesis of MASH [49].

In addition, the accumulation of mt-DNA gene mutations in patients with MAFLD was found to directly regulate mitochondrial oxidation reactions, including those encoding carrier proteins localized in the electron transport chain, peroxisome proliferator-activated receptor-γ coactivation factor 1 (PGC-1) and cytochrome P450 in OXPHOS [50]. The transcription of PGC-1 is involved in the mitochondrial antioxidant mechanism, and the accumulation of cytochrome P450 2E1 (CYP2E1) mutations affects ETC reaction activity and stimulates the accumulation of mt-ROS, resulting in an elevated trend in MAFLD [51]. In addition, it has also been shown that the induction of elevated CYP2E1 in MAFLD may be an adaptive mechanism to inhibit lipid accumulation and that CYP2E1 metabolizes FFAs in MAFLD through ω-hydroxylation. However, the hydroxylated fatty acids generated during this process are converted into cytotoxic dicarboxylic acids, which contribute to the exacerbation of MASH [52] (Figure 1).

### 2.3. Mitochondrial Quality Control

Mitochondria are highly dynamic organelles. The imbalance between mitochondrial dynamics is key to determining the total mass of mitochondrial cells in the pathogenesis of MAFLD or MASH models [53,54,55,56,57,58,59]. Abnormal expression of mitochondrial fusion proteins or giant mitochondria has been found in various mouse models of nonalcoholic fatty liver disease. Yamada, T. et al. found that excessive mitochondrial fusion or giant mitochondrial structure exists in mouse hepatocytes by treating the MASH mouse model fed a high-fat diet (MCD), although targeting Opa1 expression could effectively reduce the formation of giant mitochondria and restore the reduced succinate dehydrogenase (SDH) activity induced by MASH [60]. Du, J. et al. reported that mitochondria of hepatocytes in vivo were significantly swollen and cristae disappeared in MASH mice induced with a high-fat-high-carbohydrate-high-cholesterol diet (HFHCHCHFHCD). In the MASH model of AML-12 and HepG2 cells treated with palmitic acid (PA), the expression of mitochondrial fusion protein 1 (Mfn1) was significantly downregulated [61]. In addition, the lack of Mfn2 leads to a reduction in phosphatidyl serine (PS) transfer from the ER to mitochondria, thereby inducing ER stress and lipid accumulation. Hernandez-Alvarez, M.I. et al. found that the expression of Mfn2 significantly decreased in MAFLD patients, and in the MASH mouse model, upregulating Mfn2 alleviated the MASH phenotype [62].

In the pathogenesis of MAFLD, the dysregulation of mitochondrial dynamics is manifested by increased mitochondrial fission and decreased mitochondrial fusion. The occurrence of liver fibrosis or fat accumulation in hepatocytes and HSCs as very low-density lipoprotein (VLDL) is an important symptom of MAFLD and is regulated by mitochondrial quality control and mitophagy mechanisms. Our previous study demonstrated augmented mitochondrial fission in a CCL4-induced mouse model of liver fibrosis. Increased mitochondrial fission by overexpressing Fis1 activated HSCs, and decreased mitochondrial fission by Mdivi-1 treatment induced HSC apoptosis both in vivo and in vitro [63]. In a mouse liver fibrosis model infected with cercariae of Schistosoma japonicum, Drp1 phosphorylation at Ser637 regulated mitochondrial fission, and decreases in Opa1 and Mfn1 inhibited mitochondrial fusion and resulted in vacuolated structures [64]. Takeichi, Y. et al. found that mitochondrial fission protein Mff knockout mice were more prone to have MASH phenotypes than normal diet groups when fed a high-fat diet (HFD) [65]. Moreover, hepatocytes in Mff knockout mice had swollen balloons, and giant mitochondria increased; the expression of mitochondrial peroxisome proliferator-activated receptor-α (PPAR-α) and FAO-related protein (CPT1A) decreased significantly, and the catalyzed oxidative stress response accompanied by inflammatory cell infiltration caused upregulation of fibroblast growth factor (FGF21), which promoted the formation of MASH phenotypes.

Generally, damaged mitochondria are separated from normal mitochondria by initiating fission [33]. However, mitochondrial fission is affected by stress, and fragmentation accelerates mt-DNA fragmentation, thus stimulating the production of mt-ROS, causing oxidative stress and promoting the development of MAFLD. In addition, related studies have confirmed that Drp1-mediated mitochondrial fission is accompanied by increased levels of proinflammatory factors such as TGF-β, TNF-α, and IL-6 in MASH models. In addition, abnormal mitochondrial fission also affects autophagy pathways [63,66]. In the hepatocytes of HFD-fed Mff-KO mice, it was found that p62 and pyruvate dehydrogenase in the mitochondria increased, which inhibited the clearance of damaged mitochondria [65].

### 2.4. Mitophagy

Mitophagy is beneficial for removing damaged mitochondria and oxidative toxic byproducts (mt-ROS, dicarboxylic acid, etc.) and maintaining the normal physiological activities of mitochondria in cells.

In recent years, it has been found that mitophagy dysfunction has a nonnegligible regulatory impact on the development of MAFLD. Li, X. et al. found that the abundance of PINK and Parkin in HFD-fed mice was significantly lower than that in the control group, but the expression of P62 and LC3-I/II was significantly increased [67]. In addition, the expression of PINK1 and Parkin was significantly reduced in CCL4-induced liver fibrosis mice and Kupffer cell (KC)-transformed HSC cell models [68]. Dou, S.D. et al. reported that the mitochondrial antioxidant ubiquinone (Mito Q) alleviated the decrease in Parkin protein and restored the level of mitophagy during HSC activation [69]. In the MAFLD mouse model and PA-induced AML-12 cells, the mitophagy pathway mediated by PINK/Parkin/P62 was inhibited. Meanwhile, it was also reported that the decreased activity of peroxisome-activated receptor-γ (NR1C3) inhibited the decomposition of H_2_O_2_, induced mt-ROS-mediated cellular oxidative stress by inhibiting the PINK/Parkin pathway, and had a negative impact on the clearance of damaged mitochondria [67].

In addition, in mt-ROS-overloaded mitochondria, cytochrome c is released into the cytoplasm due to decreased ETC activity, causing apoptosis and aggravating oxidative stress. Overexpression of TIM4 upregulated mitochondrial AKT1 in KCs [68]. In addition, AKT1 could activate PTEN-induced putative kinase 3 (PINK3) to aggravate the mt-ROS-mediated BAD family (BCL2, etc.), which could bind to damaged mitochondria and induce mitophagy. Zhou, T. et al. showed that upregulation of Mst1 in primary mouse hepatocytes treated with PA could activate the expression of phosphorylated AMPK and eliminate the accumulation of damaged mitochondria caused by decreased Parkin-related mitophagy protein activity [70]. AMPK could target and inhibit the mTOR complex and activate phosphorylated ULK1, thereby eliminating the inhibitory effect of mTOR on ULK1 activation, realizing the recruitment of P62 and migrating to mitochondria, and participating in mitophagy [71,72,73].

Excessive mitophagy causes the release of mitochondrial damage-associated molecules (mt-DAMPs). Mt-DAMPs can be released from damaged mitochondria to the cytoplasm, thereby causing systemic inflammatory response syndrome to promote liver fibrosis [21,63,74,75]. In the liver, mt-DAMPs are mainly damaged by linking with TLR9 and formyl peptide receptor 1 to activate polymorphonuclear cells to promote MASH inflammation and activate HSCs, causing fibrotic scars [32,42]. However, the inhibition of mitophagy also leads to the accumulation of mt-ROS, which leads to hepatocyte necrosis and the massive release of mt-DAMPs [67]. In addition, the release of inflammatory factors and inflammasomes (NLPR3) accompanying the apoptosis of hepatocytes stimulates immune cell infiltration and HSC activation and promotes the progression of MASH [15]. NLPR3 is a multiprotein immune complex that is activated by pathogen-associated and danger-associated molecular patterns (PAMPs and DAMPs), such as lipopolysaccharide (LPS) and cholesterol (CHO) [76] (Figure 2).

“↑” represents upregulated expression; “┬” represents downregulated expression. With the stimulation of massive fat accumulation, Mfn1/2, Fis1, or Opa1 has regulatory significance for the formation of giant mitochondria. Changes such as swelling structure and vacuolization of the giant mitochondria result in the accumulation of ROS in mitochondria and HSC activation. The increased expression of phosphorylated Drp-1 and abnormal expression of Mff protein promote mitochondrial fission. A reduction in Mff inhibits the combination of PPAR-α and PGC-1, thereby reducing the synthesis and accumulation of SOD and mt-ROS. Mt-ROS can increase α-SMA, COLLAL-1, TGF-β, TNF-α, and IL-6 to aggravate inflammation and fibrosis. The combination of Parkin and PINK1 promotes the ubiquitination of mitochondrial membrane proteins, and then P62-LC-3 can target and recognize ubiquitinated membrane proteins to initiate autophagy, while downregulation of NR1C3 inhibits the Parkin/PINK1 pathway in MAFLD. At the same time, in HSCs, the increase in related immune proteins, such as TIM4, induces AKT-1 and PIK3 and the release of mt-ROS. Then, mt-ROS stimulate the BAD family and promote the release of Cyt C, causing mitochondrial oxidative stress and autophagy. PIK3 activates Parkin/PINK1. In addition, pathways such as AMPK can compensate for the abnormal mitophagy caused by the inactivation of Parkin/PINK1. AMPK controls HSC activation and mitophagy by activating phosphorylated ULK1, PPAR, and Drp-1. Upregulation of Mst1 expression can activate AMPK in MAFLD. The abbreviations in Figure 2 are defined as follows: Mfn1/2 (Mitofusion 1/2); Fis1 (Fission 1); Opa1 (optic atrophy 1); Drp-1 (dynamin-related protein 1); Mff (mitochondrial fission factor); PGC-1 (peroxisome proliferator-activated receptor gamma coactivator 1); PPAR-α/γ (peroxisome proliferator-activated receptor alpha/gamma); α-SMA (alpha-smooth muscle actin); NR1C3 (nuclear receptor subfamily 1 group C member 3); TIM4 (T-cell immunoglobulin and mucin domain-containing 4); PIK3 (phosphatidylinositol 3-kinase); AKT1 (AKT serine/threonine kinase 1); BAD (BCL2-associated agonist of cell death); Cyt C (cytochrome c); Mst (mammalian sterile 20-like kinase); AMPK (AMP-activated protein kinase); FAO (fatty acid oxidation); and ULK1 (unc-51 like autophagy activating kinase).

### 2.5. Oxidative Phosphorylation

Oxidative phosphorylation, as an important source of ATP in organisms, is the main metabolic mode of nutrient metabolism. Most of the electrons migrate down through the ETC to cytochrome c oxidase and then combine with protons and O^2−^ to form water, but there are still electrons leaking from the ETC that directly react with oxygen to form mt-ROS, such as superoxide anion (O^2−^), H_2_O_2_, etc. In recent years, mitochondrial dysfunction and abnormal production of mitochondrial ROS have been widely detected in MAFLD and MASH [77,78,79,80,81,82]. In MAFLD, activation of MPTPs and mt-DNA mutations could reduce the activity of complexes on the ETC, which is the main reason for electron leakage. Mitochondrial gene mutations, quality control, and mitophagy disorders are also important reasons for mt-ROS generation.

To maintain the balance of the mitochondrial oxidation reaction, mitochondria have antioxidation enzymes and antioxidant molecules (glutathione, VE, etc.) to eliminate mt-ROS accumulation by capturing and removing free radicals and harmful substances to repair oxidative damage [83]. Recently, it has been found that the nuclear factor-κB (NF-κB) pathway has bidirectional effects in different stages of the oxidative stress response in the pathogenesis of MAFLD [84]. In the early stage of oxidative stress, NF-κB inhibits the continuous accumulation of superoxide and accelerates the clearance of damaged hepatocytes by activating the activity of manganese superoxide dismutase (MnSOD) and the Nf-κB/JNK pathway [85]. The antioxidant mechanism of mitochondria is limited to dealing with a large number of oxidation products in the inflammatory environment because it was found that inhibitors of NF-κB kinase (IκB kinase, IKK) could be activated by phosphorylation of proinflammatory signaling molecules such as lipopolysaccharide (LPS), tumor necrosis factor (TNF), and interleukin-1 (IL-1) to inhibit the NF-κB pathway [12,86,87], accelerating the release of proinflammatory factors such as TNF-α and IL-1β, positively regulating the accumulation of mt-ROS, and eventually causing the deterioration of chronic inflammation [88,89,90]. JNK is a member of the mitogen-activated protein (MAP) kinases that participate in regulating mitophagy. In addition, the antioxidant factor PPAR coactivator-1 (PGC-1) transcription factor also has a positive effect on the recovery of mitochondrial oxidative stress [91]. In MAFLD mouse models, upregulated PGC-1α/fibroblast growth factor 21 (FGF-21) expression could restore the copy number of mt-DNA and OXPHOS activity [92]. However, in the pathogenesis of MAFLD and MASH, it was found that PGC 1α deficiency could directly downregulate superoxide dismutase (SOD), which reduced the activity of molecules such as catalase and glutathione peroxidase and further led to an imbalance of the antioxidant system [93].

The excessive accumulation of mt-ROS damages mt-DNA and combines with FFAs to participate in lipid peroxidation reactions. Moreover, mt-ROS in hepatocytes can mediate the release of TNF [16]. TNF leads to MPTP activation, ETC uncoupling, and proton pumping. In MASH mouse models and clinical cases, it was found that the byproducts formed from insufficient oxidation of overloaded FFAs, such as dicarboxylic acid, could bind with uncoupling proteins on the mitochondrial membrane, prompting protons to directly enter the matrix through uncoupling protein channels without interacting with ATP synthase, ultimately leading to a decreased ATP synthesis rate and insufficient energy production [94]. It has been found that OXPHOS driven by fatty acid β-oxidation in M2 macrophages is converted to the glycolytic pathway in the activated HSC state of mouse models of MAFLD [95]. The depletion of ATP and the increase in glycolysis activity both maintain the energy demand of cells. However, excessive glycolysis would also lead to an increase in intracellular pyruvate content and TNF or LPS, which has a partial negative effect on the lipid metabolism process of mitochondria and HSC activation.

### 2.6. Fatty Acid Oxidation (FAO)

Fatty acid oxidation is one of the main oxidation reactions that inhibits lipid accumulation in hepatocytes and provides ATP. Free fatty acids (FFAs) are substrates of various lipotoxic products, such as ceramides and diglycerides, in vivo, which induce metabolic stress and cell death [12,96]. FFAs initially decompose fatty acyl-CoA through the mitochondrial outer membrane porin and inner membrane transport protein and then participate in the tricarboxylic acid cycle reaction (TCA) to release electrons and form final products such as water to complete the consumption of liver lipids [97,98,99]. FFAs undergo esterification to form triglycerides that protect hepatocytes from lipo-toxicity at the same time.

The accumulation of lipo-toxicity induced by steatosis and excessive accumulation of FFAs are the hallmark symptoms of the development of MAFLD. Fatty acids can be catabolized in microsomes, peroxisomes, and mitochondria. In recent years, it has been found in clinical patients and animal models of MASH that the expression changes in the PPAR involved in mitochondrial fatty acid β-oxidation and regulated by mt-ROS are significantly correlated with the occurrence of MASH. Sven Francque. et al. found that in MASH patients, the expression of PPARα was significantly downregulated, and in MASH mice fed a diet lacking methionine choline (MCDD), a decrease in PPARα led to an aggravated degree of hepatitis [100]. Chen, Y. et al. found that PGC-1 was a coactivator of PPARα, and its expression was reduced with the inhibition of PPARα [101,102]. In addition, the fatty acid β-oxidation-related proteins CPT1, CPT2, ACOX1, and ACOX2 were also affected and downregulated by the downregulation of PPARα, resulting in the accumulation of lipids in cells and aggravation of the degree of hepatitis. The reduction in CPT1, CPT2, ACOX1, or ACOX2 directly inhibits the permeability of the mitochondrial membrane, preventing FFAs from entering the mitochondria for consumption and metabolism [103]. In the pathogenesis of MASH, studies have found that a lack of receptor-interacting protein kinase 3 (RIPK3) leads to the upregulation of peroxisome proliferator-activated receptor-γ (PPAR-γ) and fibroblast growth factor 21 (FGF-21) to realize the reverse recovery of MASH [47].

The other two proteins of the PPAR family, PPAR-β and PPAR-δ, are also widely expressed in hepatocytes, KCs and HSCs. In the different mouse models of MAFLD, activation of PPAR-α/β/δ induces catalase (CAT), promotes the conversion of FFAs to triglycerides to alleviate lipid toxicity, inhibits the generation of mt-ROS and the secretion of IL-1 and TNF in the lipid peroxidation reaction caused by excessive fat accumulation in mitochondria, and finally reduces the occurrence of liver fibrosis and MASH [104,105,106].

In addition, NADH and FADH2 produced by mitochondrial β-oxidation participate in ETC activities by transferring electrons. However, affected by limiting factors such as the ATP synthesis rate, electron transfer in the ETC is prone to leakage, which leads to the accumulation of mt-ROS. mt-ROS inhibit the β-oxidation of hepatic FFAs by reducing the de-palmitoylation activity of the mitochondrial outer membrane protein FAT/CD36, resulting in the accumulation of FFAs in the cytoplasm [107].

MAFLD is a polygenic disease, and the influence of gene regulation on fat oxidation metabolism is also an important factor that promotes FAO process imbalance in the development of MAFLD. For example, the nonsynonymous mutation of SOD2-C47T can reduce the activity of MnSOD and inhibit the clearance of peroxidation byproducts produced in the process of OXPHOS [108]; the mutation of UCP3-55T leads to abnormal lipolysis, causing fat accumulation and inducing inflammation [109]; GCL downregulates GCL, the rate-limiting enzyme for glutathione (GSH) synthesis, and inhibits the clearance of cellular peroxidation, resulting in the accumulation of ROS [110].

In MAFLD, it was found that in addition to the dysfunction of mitochondrial metabolism in the process of fat accumulation, organelles such as the endoplasmic reticulum (ER) also mediate the exchange of metabolites through polymeric protein complex structures such as mitochondria-associated endoplasmic reticulum membrane protein (MAM) [29,111,112,113]. When the homeostasis of the ER is imbalanced or there is a lack of energy, the ER is induced by activating the unfolded protein response (UPR), resulting in reduced GSH. The distribution of imbalance between GSH and oxidized glutathione (GSSH) causes mitochondrial stress and realizes the negative regulation of mt-ROS [21].

In brief, fatty acid β-oxidation serves as another bioenergetic pathway that occurs in mitochondria in addition to oxidative phosphorylation. Abnormalities in β-oxidation could cause the accumulation of intracellular FFAs, aggravate lipo-toxicity, and eventually induce inflammation and steatosis. In MASH patients, the accumulation of mt-ROS plays a negative role in the mitochondrial oxidation reaction and abnormal FAO, finally aggravating the accumulation of lipid substances and the formation of toxic byproducts [114] (Figure 3).

### 2.7. Gut Microbiota in the Liver–Gut Axis Influence Mitochondrial Function

The gut–liver axis is a term used to describe the interactions between the liver and resident gut microbiota in the gastrointestinal tract. Recent evidence has shown that there is a related relationship between intestinal microbiota disturbance and mitochondrial dysfunction during the development of MAFLD [115,116]. Patients with MAFLD caused by malnutrition have increased permeability of the intestinal epithelial barrier, changes in the proportion of intestinal microbial species, and bacterial translocation in the intestine, leading to abnormal liver inflammation and oxidative reactions, ultimately aggravating the development of MAFLD [24,117]. In the MAFLD mouse model, bacterial translocation, such as that of *Helicobacter pylori*, and intestinal flora metabolites (endogenous ethanol, etc.) promote the activation of neutrophils, HSCs, and Kupffer cells to produce ROS and secrete peroxidase, chemokines, proinflammatory factors, and corticosterone, thereby exacerbating hepatic steatosis and inflammation [118,119].

The gut microbiota affects the host’s synthesis of antioxidants such as glutathione. Mardinoglu, A. et al. found that the antioxidant glutathione was deficient in patients with T2D, thereby causing the accumulation of mt-ROS to aggravate oxidative stress in the liver [120]. Neish, A.S. et al. found that commensal Lactobacillus could enhance NOX family enzyme activity to promote the production of nonmitochondrial ROS, ultimately affecting the process of systemic metabolic reactions [121]. Juarez-Fernandez, M. et al. found that methylation-controlled J protein (MCJ) knockout MASH mice fed a high-fat diet had a milder degree of liver damage than WT mice. Then, through intestinal flora transplantation (FMT) into germ-free (GF) mice, they found that the proportion and formation of microorganisms and short-chain fatty acids (SCFAs) in the gut–liver axis in GF mice were improved, increasing the capacity of fatty acid oxidation in GF mice [122].

In addition, a damaged intestinal barrier is an important prerequisite for the transfer of potentially harmful bacteria and their effector molecules (PAMPs) to the liver. For example, endotoxin is a kind of LPS and a component of commensal Gram-negative bacteria. MASH patients had higher liver endotoxin levels than patients with simple steatosis [123]. Studies have found that endotoxin can mediate the NF-κB or JNK pathway to regulate cellular oxidative reactions [124]. Endotoxin can also aggravate steatosis by activating TLR and NOD-like receptors (NLRs) to produce inflammatory factors and chemokines to inhibit the decomposition of FFAs [125]. Other studies have found that excessive fructose could increase the risk of causing MAFLD, and MASH mice fed excessive fructose had an increased incidence of endotoxemia [126,127]. In fructose-treated CYP2E1-null mice, endotoxemia, inflammatory cell infiltration, and ROS production were less significantly increased than those in wild-type mice [127]. In addition, HepG2 cells treated with LPS could significantly downregulate the activities of catalase and SOD, aggravating the state of oxidative stress [128]. Above all, the relationship between oxidative stress and endotoxin regulation in MAFLD is complex. On the one hand, oxidative stress can increase intestinal permeability, leading to more endotoxin transfer into the blood and liver, and finally exacerbating liver inflammation and fibrosis. On the other hand, endotoxin can induce oxidative stress in the liver, further damaging hepatocytes by activating the NLRP3 inflammasome and releasing proinflammatory factors such as IL-1β [129].

Moreover, gut microbiota metabolites can also regulate mitochondrial structure. For instance, Zhao, T. et al. found that butyrate, an intestinal flora metabolite, can significantly alleviate mt-ROS copies caused by a high-sugar diet in db/db mice, reduce their number and content, and effectively restore mitochondrial membrane potential and ATP synthesis capacity [130]. Previous studies have shown that Bacteroidetes and Firmicutes promote an increase in the level of SCFAs, which are used by mitochondria to synthesize ATP [117]. Butyrate can also reverse the decrease in the activity of fatty acid β-oxidation rate-limiting enzymes such as CPT1A and acetyl-CoA carboxylase α (ACACα) to improve fatty acid metabolism and thereby reduce the degree of hepatocellular hypertrophy and steatosis [130].

## 3. Treatment Approaches for Mitochondrial Dysfunction-Related Oxidative Stress in MAFLD

In recent years, the prevalence of MAFLD has been increasing year by year according to the proportion of regions. Through the statistics of the current clinical research on MAFLD, it was found that there is medicine that can target metabolic processes in different ways to alleviate MAFLD.

### 3.1. Antioxidant Trace Elements

By comparing the serum of MAFLD and MASH patients with that of normal people, it was found that the expression of vitamins in the serum showed a significant downward trend [113,131]. According to a number of experimental models, injecting vitamins can significantly improve the occurrence of inflammation, liver fibrosis, and the worsening trend of MAFLD. The vitamin family is an important class of antioxidant molecules in the body that regulate oxidative stress and fat accumulation by catalyzing enzymatic reactions.

#### 3.1.1. Vitamin C

Vitamin C (VC) is a common water-soluble molecule, and we often supplement the VC content in the body in the form of an aqueous solution daily. According to its antioxidant properties, VC plays an important role in scavenging oxygen free radicals. The basic mechanism by which VC alleviates the degree of oxidative stress is through reducing the generation of mt-ROS and increasing the levels of antioxidant enzymes such as superoxide dismutase and GSH peroxidase, thereby improving the activity of the ETC [83]. At the same time, VC inhibits fat accumulation by activating PPARα in MAFLD. High-dose VC stimulates the downregulation of adiponectin, which has inhibitory effects on lipid accumulation, IR, and inflammation in the liver [132].

Clinical investigations have shown that the lack of VC in the body (VC deficiency) successively causes liver fibrosis and MAFLD aggravation, especially in obese patients or overweight patients (BMI > 24) [133]. In the mouse model of a high-fat diet, appropriately supplementing a large amount of VC (beyond what is necessary for the body) would effectively alleviate the fat accumulation and MAFLD caused by high sugar and fat. At the same time, different concentrations of VC were used to treat MAFLD, and it was found that supplementation with low concentrations had positive significance in preventing liver inflammatory diseases in normal mice [132,134]. In addition, it was confirmed in a guinea pig model that VC deficiency has no significant effect on the development of MAFLD under a low-fat diet, but if a high amount of VC was combined with a low-fat diet, it obviously promoted the development of liver inflammatory disease reversal in healthy liver [135]. Even though in different animal models the intake of microbial C has different significance for the occurrence of MAFLD, the conclusion of the comprehensive experiment found that VC is indeed a good target for clinical intervention of liver inflammatory diseases. However, its specific function in the human body and therapeutic drug doses still need further investigation.

#### 3.1.2. Vitamin E

Vitamin (VE) is an important antioxidant inhibitor against the conversion of simple steatosis to MASH caused by excessive oxidative stress. Similar to the role of VC, the detoxification function of VE is mainly through capturing and providing electrons for free hydroxyl radicals and H_2_O_2_ and then combining with antioxidant enzymes to detoxify them into water and oxygen [136]. VE also has potential effects on the development of inflammation. VE downregulates the expression of trans growth factor-β (TGF-β) through mt-ROS [136], resulting in reduced forms of nitric oxide synthase and NADPH oxidase by reducing the frequency of oxidative stress to affect the development of liver fibrosis [137]. In addition, it was found in hepatitis mice that feeding with an MCD diet can restore glutathione and significantly reduce the expression levels of oxidative stress markers, HSC activation, and fibrosis under VE injection [138,139]. VE can also activate PPARα transcription, promote the action of the adiponectin promoter, and enhance IR, thereby alleviating the development of MAFLD [140].

Due to the active physiological functions of VE, it is also recognized as an effective targeted therapy drug in clinical treatment. In data collected through broad-spectrum surveys of the population, it was found that VE tended to show more obvious light fibrosis symptoms than control groups [141]. In addition, the results showed that in children with nonalcoholic liver disease, compared with the injection of a placebo, hydroxytyrosol and vitamin E significantly improved the degree of steatosis and weakened the occurrence of systemic inflammation by promoting the systemic circulation of the anti-inflammatory factor interleukin-10 [13]. In conclusion, VE is widely used in combination with other biological drugs to regulate the level of hepatic oxidative stress and FAO, but its specific appropriate doses for the clinical treatment of MAFLD and MASH still need to be further determined and explored for clinical applications.

#### 3.1.3. Vitamin D

Vitamin D (VD) is a multifunctional hormone that not only stabilizes calcium homeostasis and regulates bone mineralization but also plays an important role in the regulation of immunity and inflammation. VD deficiency has been associated with the development of metabolic syndrome-associated diseases such as T2DM and MAFLD [142]. In a study of FFA decomposition in MAFLD models, VD acted on outer membrane lipoprotein through the PPARα/CPT1A pathway to catalyze the mitochondrial β-oxidation process in hepatocytes and inhibited the denaturation of lipoproteins and reduced the development of MAFLD [143].

VD also controls the endoplasmic reticulum–mitochondrial stress-oxidative response by promoting the nuclear translocation of the antioxidant molecule nuclear factor erythroid 2-associated factor 2 (NFE2L2), reducing Toll-like receptor expression [144,145]. In addition, a study showed that VD mediated the activation of hepatocyte nuclear factor 4α (HNF4α) through specific receptors to improve hepatic insulin resistance and reduce the possibility of hepatic steatosis [143] At present, the effect of VD on the clinical treatment of chronic inflammatory liver diseases is still unstable, and the impacts of the intake of different concentrations of VD on health are worthy of further research.

#### 3.1.4. Vitamin A

Vitamin A (VA) is a necessary regulatory factor for many physiological processes, such as visual perception, cell proliferation and differentiation, immune response, and metabolic regulation. Retinoic acid (retinol), a metabolite of vitamin A, plays an important role in the function of VA. Retinoic acid receptor (RAR) and retinoid receptor (RXR) jointly control the activation of retinoic acid transcription factors. The RAR and RXR receptors, which are the unique structure of vitamin A, can also affect the process of fat oxidation by acting on the PPARα pathway [146].

Nonactivated HSCs in the liver are the main storage location of VA in the body, which also reveals that VA deficiency is inextricably linked to the occurrence of liver fibrosis [147]. According to the data collected in existing clinical research from MAFLD patients, VA metabolism disorder is not only manifested in the downregulation of overall expression in serum but also shows an imbalance in the ratio of retinal lipids and retinol, as well as a change in intracellular retinoid storage locations, such as retinol transferred from HSCs to hepatocytes [103]. However, summarizing recent research, it has been found that the treatment of chronic liver inflammatory diseases by directly increasing vitamin A intake is not stable for the treatment of MAFLD. This may be related to the individual differences in patients and related molecular mechanisms, such as retinoid lipid overload accumulation in hepatocytes, which also provides a new direction for further exploration of VA to improve HSC activation and FFA accumulation in MAFLD.

#### 3.1.5. Vitamin B

In the past decade, several vitamin B subfamilies have been studied in MAFLD conditions in existing clinical studies.

Vitamin B3 (Niacin) is the precursor molecule of the coenzymes nicotinamide adenine dinucleotide (NAD) and nicotinamide adenine dinucleotide phosphate (NADPH), which regulate various physiological responses [133,148]. In a rat model and a high-fat HepG2 model, increasing niacin restored the loss of mitochondrial redox potential, ROS accumulation, lipid digestion in the liver, and NADPH enzyme activity reduction in chronic inflammation [149,150].

Vitamin B9 (folic acid) and vitamin B12 (cyanocobalamin) are both related to the occurrence of MAFLD-related comorbidities such as obesity and T2DM. Folate acid protects the liver by restoring activation of adenosine monophosphate-activated protein kinase (AMPK). Vitamin B12 is the co-factor of methylmalonyl-CoA mutant enzymes that affect DNA synthesis repair and mitochondrial metabolic homeostasis, which are involved in the pathogenesis of MAFLD [151]. However, according to current clinical case data, there are not enough complete and sufficient results to explain whether the injection of the vitamin B family regulates the occurrence of MAFLD [152,153].

#### 3.1.6. Coenzyme Q

Coenzyme Q (CoQ) is a self-synthesized fat-soluble bioactive quinone, similar to vitamin E. CoQ is located on the inner membrane of mitochondria and is widely distributed in various tissue cells. The function of CoQ is mainly through participating in mitochondrial ETM, improving electron transport efficiency, and maintaining redox homeostasis by utilizing the structural transformation between the three redox forms of ubiquinone, semi-ubiquinone, and ubiquinol, and directly acting on free radicals and oxides to restore cell viability [154].

Clinical data have shown that CoQ10 supplements can alleviate the oxidation level of type I, II, and III complexes in the respiratory chain, regulating OXPHOS and inhibiting mitochondrial dysfunction induced by MAFLD [155]. Alhusaini, A.M. et al. found that liposome-encapsulated coenzyme Q could significantly reduce liver damage and fibrosis caused by propionic acid by inhibiting cytochrome C and mitochondrial fragmentation and simultaneously increasing the expression of Bcl-2 [156]. Tiefenbach, J. et al. proposed that coenzyme Q and its analog Idebenone can act as PPARα/γ agonists and downregulate triglyceride and cholesterol levels, finally reducing the development of steatosis and MAFLD [157]. Sumbalova, Z. et al. used hydrogen-rich water (HRW) to treat MAFLD clinical patients and mouse models and found that HRW has the potential to help MAFLD patients restore normal coenzyme Q expression levels and mitochondrial oxidation function, ultimately realizing the potential of MAFLD treatment [158]. In mouse MAFLD models, very-low-density lipoprotein (VLDL) was overproduced in a high-fat diet. CoQ10 was shown to guide VLDL to accumulate and transform into a larger volume so that it is more easily recognized and degraded by enzymes to control the occurrence of lipid peroxidation and oxidative stress [159]. Notably, a large volume of VLDL also causes other vascular diseases, such as arteriosclerosis and other side effects. Hence, it is important to solve the main question about the by-effect in the process of treating MAFLD for CoQ through clinical treatment or animal models.

### 3.2. Nrf2-Antioxidant Supplement

Nuclear factor erythroid 2-related factor 2 (Nrf2) is a high-affinity electron-competent transcription factor involved in antioxidant effects. Nrf2 can inhibit excessive oxidative reactions such as endoplasmic reticulum stress by regulating downstream antioxidative stress genes such as heme oxygenase-1 (HO-1) and superoxide dismutase (SOD). Due to its outstanding antioxidant capacity, there are currently a variety of Nrf2 supplements used in the disease research of MAFLD and MASH [87].

#### 3.2.1. Aucubin

Aucubin (AU) is a natural compound that can be extracted from plants and has anti-inflammatory effects. AU relieves lipid accumulation by promoting the expression of Nrf2 and PPAR in mice and 3T3-L1 cells (a type of macrophage) and inhibits the release of proinflammatory cytokines such as TNF-α, IL-1β, and IL-6, in turn reducing oxidative stress and the inflammatory response by enhancing the oxidative stress and AMPK/AKT phosphorylation associated with hyperlipidemia [160]. AU could counteract the hepatic fibrosis caused by CCL4 and α-amanitin, showing therapeutic significance for the treatment of MAFLD.

#### 3.2.2. Melatonin

Melatonin is a small molecule indole amine substance that is mainly related to the regulation of biological rhythm in animals. Melatonin has antioxidant properties and counteracts the negative effects of active oxygen in the body. Joshi, A. et al. found that melatonin-mediated HepG2 cells reduce oleic acid uptake and increase mitochondrial membrane potential. In high-fat-diet-fed mice treated with melatonin, Nrf2/HO-1 activity was restored, and the expression of Nrf2-Keap1 in hepatocellular mitochondria was increased, reducing intracellular oxidative stress levels and alleviating MAFLD [161].

Research on drugs targeting the Nrf2/OH-1/Keap1 pathway, intracellular oxidation reactions, and lipid degradation processes has become a new hot topic. In addition to AU and melatonin, a variety of natural compounds have been used in the prevention of MAFLD/MASH and have obvious treatment effects. For example, inhibition of the upregulation of p62 transcription regulated by ROS/P38/Nrf2 alleviated the oxidative stress damage of macrophages caused by LPS [58,162]. Naringin downregulates the expression of the targeted proteins ChREBP, SREBP-1c, nSREBP-1c, ACC, and FAS to inhibit the accumulation of fatty acids and regulate Nrf2-HO-1/Nf-κB to reduce the impact of inflammation on the development of MAFLD [163].

### 3.3. MicroRNA

MicroRNA (miRNA), as an important bioactive factor target, has been widely found and studied in specific epigenetic mechanisms. In recent years, abnormal expression of miRNAs has been found in metabolic disorders such as MAFLD and MASH [164,165,166]. For example, it was found that miRNA-21/20B was upregulated in the inflammatory environment in the HFD-fed mouse model. Mt-ROS and nitric oxide (NO) generated from activated HSCs could target miRNA-21 to form more pro-fibrotic proteins, such as type I collagen (Col1α1) and α-SMA [136]. MiRNA20B in MAFLD mediates PPAR-α activity to reduce the occurrence of abnormal FFA oxidation and mitochondrial dysfunction [167]. In addition, circRNA-002581 inhibits the expression of miRNA-122, which promotes fat oxidation and the activation and phosphorylation of mTOR to aggravate the development of inflammation [168].

Other studies confirmed that miRNA-34a and miRNA-223 inhibited the Sirtuin 1/AMPK/PPARα pathway through vesicle enrichment [169,170]. MiRNAs play a therapeutic role in the process of chronic inflammation and oxidative stress. As a result, miRNAs have become new treatments to eliminate the development of MAFLD and MASH.

### 3.4. Targeted Microbiota Therapies Targeting the Liver–Gut Axis

#### 3.4.1. Gut *Akkermansia muciniphila*

In recent years, beneficial microorganisms such as *Akkermansia muciniphila* have been considered promising research hotspots in combating MAFLD. Morrison, M.C. et al. used heated *Myxiniphila* to treat HFD-induced obese mice and found that it could significantly decrease intestinal permeability and adipocyte hypertrophy [171]. Rao, Y. et al. treated MAFLD mice with oral administration of Proteobacillus *Myxophila* and found that mt-DNA copy number and oxidative metabolism markers such as PGC-1α and CPT-1β increased in the hepatocytes of the treatment group. In addition, the expression of mitochondrial complexes I–V was significantly upregulated. In addition, *Myxophila* stimulated L-aspartic acid, thereby activating AMPK activity and sustaining lipid oxidation reactions, effectively inhibiting liver lipid accumulation [25]. At this stage, various research data show that whether it is through direct oral administration or through dietary or drug intervention in the MAFLD mouse model, the abundance of *Myxophila* in the body shows great therapeutic potential for the alleviation of MAFLD. However, clinical efficacy verification in other animals is still lacking, and the reduction in therapeutic efficacy after inactivation needs further investigation [172,173].

#### 3.4.2. Bile Acids (BAs) and Short-Chain Fatty Acids (SCFAs)

Intestinal metabolites such as bile acids (BAs) and short-chain fatty acids (SCFAs) are important molecules involved in the enterohepatic circulation and metabolism of the body, affecting the development of MAFLD. BAs regulate lipid metabolism, gluconeogenesis, and ATP synthesis processes by activating Farnesoid X receptor (FXR), the G protein-coupled receptor superfamily (TGR5), and other receptors that are highly expressed in the liver and intestines [174]. The FXR-TGR5 dual agonist INT-767-treated Western diet (WD)-fed mice could reduce fatty acid synthesis and alleviate AMPK, SIRT1/SIRT3 phosphorylation, and mitochondrial complex IV activity, decreasing trends caused by WD [175]. However, excessive activation of FXR can lead to an increase in total cholesterol and low-density lipoprotein cholesterol levels in MASH patients; therefore, the safe dosage of FXR agonists still needs further prediction and evaluation.

Short-chain fatty acids (SCFAs) are a class of saturated and fatty acids produced by intestinal flora through fermentation of soluble dietary fiber, including acetate, propionate, and butyrate. Zhao, T. et al. found that butyrate supplementation could inhibit the activity of NADH oxidase in the ETC of MAFLD patients, increase the concentration of potassium ions in the mitochondria, maintain the mitochondrial membrane potential, and delay the development of MAFLD [130,176]. Acetate and propionate mainly maintain the intestinal barrier and reduce the release of inflammatory factors such as IL-6 [177].

### 3.5. Other Molecular Drugs

Several experimental targets are being explored in clinical practice to delay the sustained development of MASH, including FXR agonists, FGF21, glucagon-like peptide-1 (GLP-1), PPAR agonists, etc. GLP-1 is an incretin hormone that regulates blood sugar and weight homeostasis. Clinical diabetic patients with low expression of GLP-1 receptor (GLP-1R) are known to have mt-ROS accumulation, superoxide formation, and membrane potential loss [178]. The GLP-1 analogs liraglutide and semaglutide have been proven to delay the development of MASH. Semaglutide can reduce the occurrence of mitochondrial dysfunction by enhancing autophagy and resisting oxidative stress in neurodegenerative diseases [179]. During the treatment of MASH patients, semaglutide can also bind with GLP-1R to effectively improve glucose–lipid metabolism and reduce oxidative stress in hepatocytes [180]. Liraglutide increases the levels of mitochondria-related structural proteins, such as Drp1, OPA1, and UCP2, enhances mitochondrial structural remodeling, and restores the expression of autophagy-related proteins, such as Beclin 1, LC 3, and Sirtuin 1, to inhibit the development of MAFLD [181]. The combined use of multiple drugs has shown a stronger effect on the control of MAFLD than single drugs. For example, the combined use of FXR agonist (Cilofexor), ACC inhibitor (Firsocostat), and semaglutide in MASH patients was compared with semaglutide alone, and the combined use of the drugs improved the two indicators of transaminase and fat content more significantly [182] (Table 2).

## 4. Summary

At present, MAFLD has become a global health and safety issue by affecting a quarter of the population in the world and has no approved drug therapy. Furthermore, in recent years, studies have shown that MAFLD is inextricably linked to the development of extrahepatic diseases. Tarantino, G. et al. used the triglyceride/glucose (TyG) index and triglyceride/high-density lipoprotein (HDL) to detect the occurrence of MAFLD and IR, respectively, in bladder cancer patients and found that both TyG and HDL significantly increased [183]. Liang, Y. et al. used 6873 Chinese middle-aged and elderly people as clinical research subjects and found that patients with MAFLD had a significant increase in the risk of cardiovascular diseases by inducing abnormal elevation of arterial hyperlipidemia and causing myocardial damage [184,185]. At the same time, liver fibrosis inhibits the filtering function of glomeruli to injure kidney functions [186]. In addition, for MAFLD patients of different sexes, the risk of colon cancer in men and breast cancer in women was also significantly increased. Mantovani et al. conducted a systematic summary of the association between MAFLD and extrahepatic cancers through follow-up observation statistics of clinical patients and found that MAFLD increased the risk of extrahepatic cancer, including increasing the risk of developing gastrointestinal cancers, such as esophageal, gastric, pancreatic, and colorectal cancers, by nearly 1.5- to 2-fold [186,187,188]. In addition, MAFLD can also significantly increase the probability of breast cancer, thyroid disease, etc. [189]. However, it is worth noting that the probability of extrahepatic cancers increased by MAFLD is affected by a variety of confounding factors, including age, sex, smoking, obesity, and other potential factors, so these related mechanisms need to be further researched.

As important organs for energy formation and metabolism in cells, mitochondria mainly metabolize carbohydrates and fatty acids through oxidative phosphorylation and β-oxidation, respectively, in hepatocytes. However, when the mitochondrial structure is damaged, the genome expression is abnormal, or the quality control is disordered, which leads to the collapse of ETC, the reduction in ATP synthesis, and the massive accumulation of mt-ROS and intermediate metabolites of fatty acid decomposition such as dicarboxylic acid. This review mainly focuses on the causes and impacts of mitochondrial dysfunction on the pathogenesis of MAFLD. In addition, there are potential therapeutic effects of natural antioxidant compounds and molecular medicine to treat MAFLD.

In addition, Fu, A. et al. used tail vein injection of healthy mitochondria into MAFLD mice to compensate for the mitochondrial physiological activity of missing functions. This approach effectively reduced serum aminotransferase activity and blood lipid content and significantly restored ATP synthesis, cytochrome oxidase activity, and mitochondrial antioxidant formation [163]. At present, there is still a lack of other animal models or clinical cases to further determine the therapeutic effect of mitochondrial reinfusion, while it is undeniable that it will be a new potential strategy for the treatment of MAFLD.

The pathogenesis of MAFLD or MASH is highly heterogeneous and influenced by different triggers at different ages. For example, simple steatosis induced by genetics, malnutrition, or changes in the environment eventually leads to MAFLD, which is the main cause of illness in children and adolescents. In adults, there are many causes, including non-excessive alcohol, genetics, diabetes, etc. These causes can also trigger mitochondrial dysfunction. In addition, the manifestations of mitochondrial dysfunction vary among different disease types. Comprehensive clinical data analysis revealed that the number of mitochondria in the liver of patients with MASH is reduced, and the structure is more obviously damaged than that in normal people, which in turn aggravates the process of liver cell necrosis and fibrosis.

Clinical data and research provide evidence to support that mitochondrial dysfunction influences the development of MAFLD by changing oxidation reactions and antioxidant mechanisms. We have provided a simple table to summarize these main therapeutic strategies for mitochondrial function in MAFLD or MASH in this review. However, mitochondrial dysfunction in MAFLD has yet to be completely elucidated. Therefore, there is virtual significance to further determine different types of antioxidant molecules in the occurrence of MAFLD. It is equally important to determine whether the use of the above antioxidants would cause side effects and impact the treatment effect at different stages of MAFLD or MASH. Risk assessment and prediction in clinical treatments for MAFLD and MASH patients through transplantation of beneficial intestinal flora and mitochondrial reinfusion also require further verification.

## Figures and Tables

**Figure 1 ijms-24-17514-f001:**
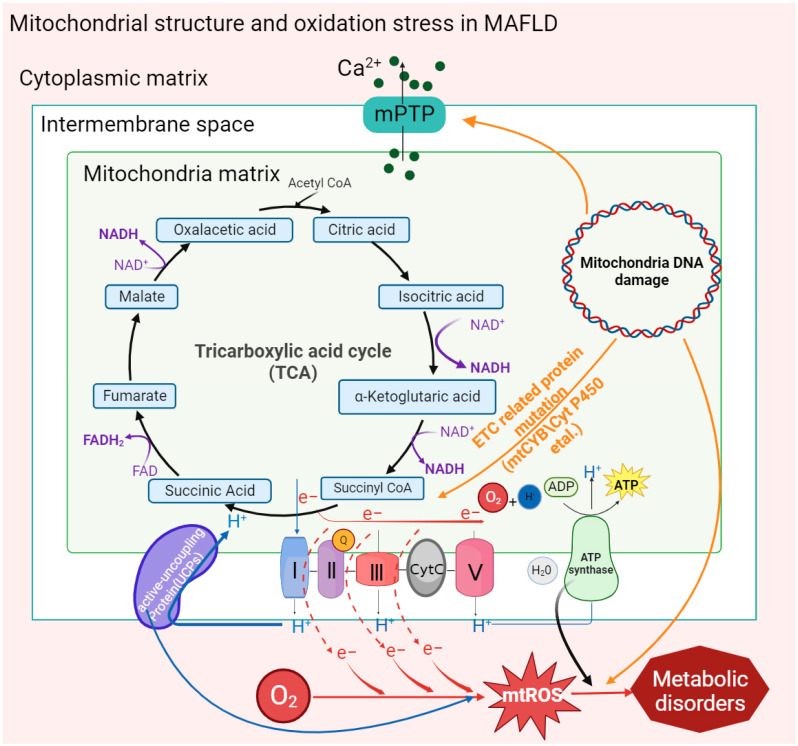
Oxidative stress caused by mitochondrial structures and mt-DNA mutations in MAFLD. Activation of the mitochondrial membrane permeability transition pore (MPTP) by factors such as mitochondrial mutant genes and the accumulation of fatty acids promotes the outflow of Ca^2+^ from the mitochondrial calcium pool and then stimulates the activity of inner membrane proteins to affect the ATP synthesis rate and form new transition pores. Mitochondrial gene mutation (mt-DNA) also stimulates the activation of MPTP and uncoupling proteins (UCPs). In addition, the reduction in the activity of inner membrane proteins leads to electron leakage in the ETC, which promotes the generation of mt-ROS and ultimately aggravates the degree of mitochondrial oxidative stress in MAFLD.

**Figure 2 ijms-24-17514-f002:**
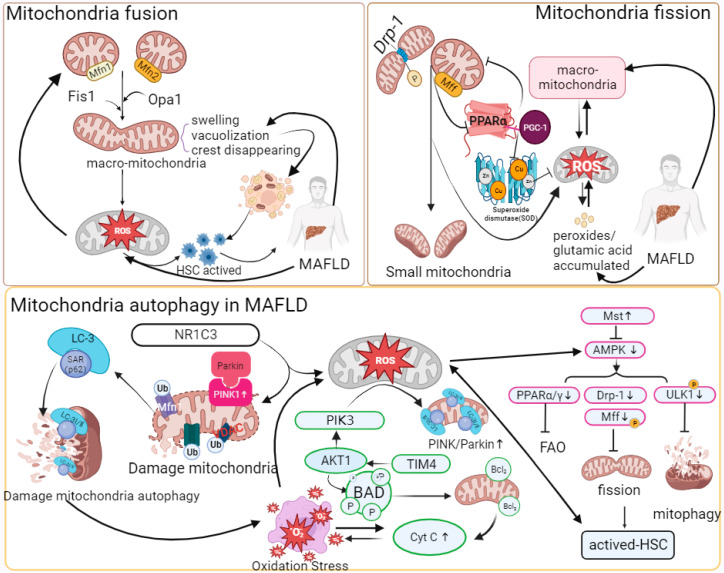
Mitochondrial quality control and oxidative stress in MAFLD.

**Figure 3 ijms-24-17514-f003:**
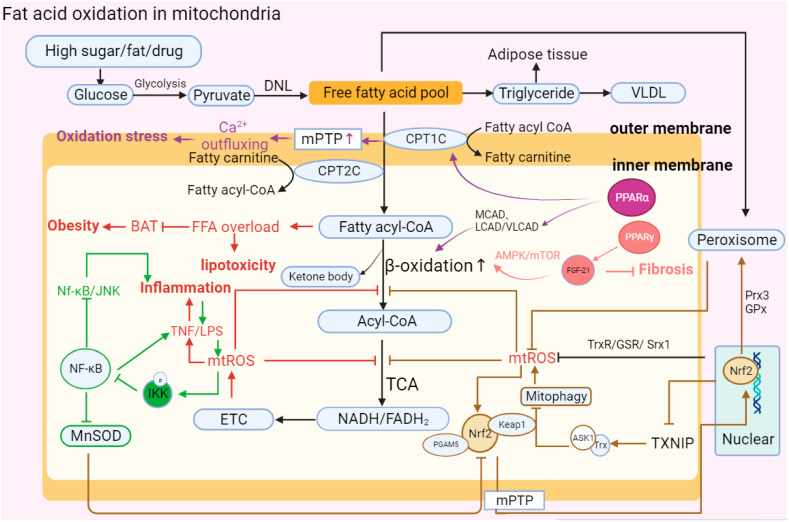
Abnormal fatty acid oxidation in mitochondria. “↑” represents upregulated expression; “┬” represents downregulated expression. Carbohydrates in hepatocytes are initially metabolized into pyruvate, and then pyruvate forms free fatty acids (FFAs) through DNL. Finally, FFAs form triglycerides and VLDL in MAFLD. Upon stimulation such as the introduction of high sugar and fat, the activation of MPTP and decrease in PPARα leads to the abnormal structure of CPT1C and CPT2C, which transport FFAs on the mitochondrial membrane. FFAs flow into the mitochondrial matrix, causing lipotoxic accumulation and further affecting BAT consumption, which leads to obesity. FFAs in the matrix are decomposed into acyl-CoA via β-oxidation and then participate in oxidative phosphorylation. In MAFLD, PPARα can reduce the activity of β-oxidase and inhibit the decomposition of FFAs, while increasing the activity of PPAR-γ can promote the expression of FGF-21 and restore the oxidation of FFAs through the AMPK-mTOR pathway to inhibit the development of fibrosis. Moreover, mitochondrial oxidation imbalance causes the accumulation of mt-ROS, TNF, and LPS, aggravating the development of inflammation. At the same time, TNF can activate phosphorylated IKK to inhibit the NK-κB antioxidant response, including promoting the continuous accumulation of superoxide by reducing MnSOD activity and the Nf-κB/JNK reaction. In addition, Nrf2 mediates oxidation reaction efficiency and participates in fat metabolism. Nrf2 forms a ternary complex with Keap1 and PGAM5 to directly respond to mt-ROS and stimulates nuclear Nrf2 to be accelerated and transported into mitochondria. Nrf2 can eliminate excess mt-ROS accumulation and restore β-oxidation by enhancing the expression of Trx, GSR, Srx1, Prx3, and GPx. However, Nrf2 also participates in regulating the occurrence of mitophagy and aggravating ROS accumulation by reducing the expression of TXNIP and increasing the combination of Trx and ASK1.The abbreviations in Figure 3 are defined as follows: DNL (de novo lipogenesis); VLDL (very-low-density lipoprotein); MPTP (mitochondrial permeability transition pore); CPT1C (carnitine palmitoyl transferase 1C); CPT2C (carnitine palmitoyl transferase 2C); MCAD (medium-chain acyl-CoA dehydrogenase); LCAD (long-chain acyl-CoA dehydrogenase); VLCAD (very-long-chain acyl-CoA dehydrogenase); AMPT/MTOR (adenosine monophosphate-activated protein kinase/mammalian target of rapamycin); BAT (brown adipose tissue); NK-κB (nuclear factor kappa-light-chain-enhancer of activated B cells); LPS (lipopolysaccharide); IKK (IκB kinase); MnSOD (manganese superoxide dismutase); ETC (electron transport chain); TCA (tricarboxylic acid cycle): NADH/FADH2 (nicotinamide adenine dinucleotide (reduced form)/flavin adenine dinucleotide (reduced form)); Nrf2 (nuclear factor erythroid 2-related factor 2); PGAM5 (phosphoglycerate mutase family member 5); Prx3 (peroxiredoxin 3); TXNIP (thioredoxin-interacting protein); and ASK1 (apoptosis signal-regulating kinase).

**Table 1 ijms-24-17514-t001:** Mitochondrial functions in different chronic liver diseases.

	Steatotic Liver Disease (SLD) [26,27,28]	Metabolic Dysfunction-Associated SLD (MASLD/MAFLD) [29,30,31,32,33,34]	Excessive Alcohol and Metabolic-Associated SLD (MetALD) [35,36,37]	Drug-Induced Liver Injury (DILI) [38,39,40]
Mitochondrial structure	The electron transport chain (ETC) is disrupted; the activity of mitochondrial complex III is decreased.	Mitochondrial membrane permeability increases; cristae disappear and giant mitochondria appear; and membrane potential decreases.	Mitochondria swell; mitochondrial membranes rupture; and membrane potential disappears.	Mitochondrial outer membrane is damaged; mitochondrial membrane potential decreases.
Energy metabolism	ATP synthesis is inhibited; the TCA cycle is disrupted.	ATP synthesis is reduced; oxidative phosphorylation and fatty acid oxidation efficiency are reduced.	ATP synthesis is downregulated.	ATP deficiency; ETC is damaged; and succinic acid is accumulated.
Mitochondrial DNA (mt-DNA)	mt-DNA content and mitochondrial density increases.	The fragmentation of mt- DNA is increased; the frequency of mitochondrial mutations is increased.	The fragmentation of mt-DNA is increased.	The copy number of mitochondrial is reduced and mt-DNA is depleted.
Mitophagy	Mitophagy is reduced.	Mitophagy is reduced; mitochondrial quality control homeostasis is imbalanced; and mitochondrial fission is increased.	Excessive mitochondrial autophagy.	Mitochondrial selective autophagy is reduced.
mt-ROS	mt-ROS is increased, which is caused by incomplete oxidation of substrates such as succinic acid.	The increase in mt-ROS production is affected by diet, lifestyle, genes, etc.	mt-ROS increases via fat accumulation, etc.	mt-ROS increases.

**Table 2 ijms-24-17514-t002:** The main therapeutic strategies for mitochondrial function in MAFLD/MASH.

Name	Pathway	Treatment	Model	Effect for MAFLD
Antioxidant trace elements				
Vitamin C (VC)	mt-ROS↓, adiponectin↓; PPARα↑, antioxidant enzymes↑	High-dose intake	Mouse (High-fat diet)	Reduce lipid accumulation, IR, and inflammation
		High-dose intake + low-fat diet	Guinea pig	
Vitamin E (VE)	Mt-ROS↓, iNOS↓, NADPH oxidase↓; PPARα↑	Oral administration of hydroxytyrosol and VE	MAFLD children	Reduce HSC activation and fibrosis
Vitamin D (VD)	mTOR↓, Sirtuin↓; PPARα/CPT1A↑, HNF4α↑, NFE2L2↑	Gavage dose of VD	Wistar rats	Reduce liver steatosis, serum lipid accumulation
		VD	HepG2 cell by OA	Inhibit lipid and TG accumulation in cell
Vitamin A (VA)	PPARα↑, FGF21↑, CPT1A↑, UCP2↑	——	——	VA deficiency in MAFLD patients
Vitamin B (VB)	ROS↓; restore lipid digestion, the activity of NADPH enzyme, and mitochondrial redox potential	0.5% niacin in the diet	Rat (High-fat diet)	Reduce chronic inflammation and hepatic steatosis
Coenzyme Q				
L-CoQ10	Restore OXPHOS and mitophagy; increase the activity of cytochrome C	Oral administration L-CoQ10	Rat (orally intoxicated with PRA-induced MASH)	Reduce liver damage and fibrosis
Idebenone	PPARα/γ↑; triglyceride↓, cholesterol↓	Oral administration Idebenone	Mouse model of type 2 diabetes	Reduce hepatic steatosis
Hydrogen-rich water	Restore normal coenzyme Q expression levels	Oral administration	MAFLDH patients/mouse models	Potential therapy to alleviate MAFLD
Nrf2 antioxidant supplement				
Aucubin	Nrf2↑, PPAR↑, p-AMPK/AKT↑; TNF-α↓, IL-1β↓, IL-6↓	Intraperitoneal injection of aucubin	Mouse (Tyloxapol-induced MAFLD)	Reduce lipid accumulation, oxidate stress, and inflammation
Melatonin	Mt-ROS↓; restore the activity of Nrf2/HO-1 and mitochondrial redox potential	Intraperitoneal injection of melatonin	Mouse (High-fat diet)	Reduce oxidate stress and damage to hepatocytes
		——	HepG2 by OA	
Scoparone	Mt-ROS/P38/Nrf2↓, P62↓	——	Macrophages by LPS	Alleviate oxidative stress damage
MicroRNA				
miRNA21/20B	——	——	Mouse (CCL4-induced MAFLD)	Upregulate in MASH mice
miRNA-223	Vesicle enrichment to inhibit Sirtuin 1 and AMPK activation; mt-ROS↓; Nrf2↑, HO-1↑, SOD1/2↑	miR-223 expressed by elevated EA	HepG2 cells (High glucose-induced MAFLD)	Reduce oxidative stress and insulin resistance
Microbiota and intestinal metabolites				
*Akkermansia muciniphila*	Restore mt-DNA, PGC-1α, CPT-1β, and activate AMPK activity	Oral administration of *Akkermansia muciniphila*	Mouse (HFD diet)	Reduce liver lipid accumulation
Bile acids	Activate Farnesoid X receptor (FXR), G protein-coupled receptor superfamily (TGR5), and ATP synthesis processes	Intraperitoneal injection of INT-767	Mouse (HFD diet)	Reduce fatty acid synthesis, AMPK, SIRT1/SIRT3 phosphorylation, and the decreasing trend of mitochondrial function
SCFAs	Maintain the mitochondrial membrane potential	Butyrate supplementation	MAFLD patients	Delay the development of MAFLD
Other molecular drugs and strategies				
Semaglutide	Bind with GLP-1R to decrease the accumulation of mt-ROS, superoxide formation, and membrane potential loss	——	MASH patients	Improve IR and glucose–lipid metabolism, and reduce oxidative stress
			SH-SY5Y cell	
Liraglutide	Drp1↑, OPA1↑, UCP2↑, Beclin1↑, LC3↑	Subcutaneous injection of liraglutide	Mouse (HFD diet)	Inhibit the development of MAFLD

“↑” represents upregulated expression; “↓” represents downregulated expression; “——” represents unknown. The abbreviations in Table 2 are defined as follows: L-CoQ10(Liposomal-coenzyme Q10); SCFAs (short-chain fatty acids); PPARα (proliferator-activated receptor-alpha); FGF21 (fibroblast growth factor 21); CPT1(Carnitine Palmitoyl Transferase I); UCP2(Uncoupling Protein 2); NFE2L2 (Nuclear Factor Erythroid 2-associated Factor 2); HNF4α (Hepatocyte Nuclear Factor 4α); HFD-diet (high-fat diet); EA (ellagic acid); INT-767(FXR-TGR5 dual agonist).

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
