# Peer review of "Mitochondrial Dysfunction in Metabolic Dysfunction Fatty Liver Disease (MAFLD)"

_ijms, 2023, doi:10.3390/ijms242417514_

Round 1

Reviewer 1 Report

Comments and Suggestions for Authors

In recent decades, MASLD has emerged as one of the most prevalent liver diseases worldwide, in fact it has become a major global health care concern. It is due to the fact that the mechanisms involved in the development of the disease are unknown as well as the lack of approved therapeutic options for the treatment of MASLD. The aim of this literature review is to highlight the role of mitochondrial dysfunction in the development of this disease, in addition to describing possible therapeutic strategies to be addressed in the coming years using mitochondrial dysfunction and oxidative stress as targets. I appreciate very much that the authors have detailed the major structural as well as metabolic modifications associated with mitochondrial dysfunction that are implicated in the development of MASLD. However, I think it is necessary to consider including some issues in the manuscript to achieve an updated and innovative review.

1.     I strongly believe that the review requires a section about the interaction between mitochondrial dysfunction and the gut-liver axis. Indeed, due to the relevance of mitochondria-microbiota interaction during NASH, the authors should include the strategy as a potentially promising therapeutic approach to be addressed in the coming years (Juarez-Fernandez M. et al. Hepatology 2023).

2.     The authors' effort to detail the structural mechanism of mitochondrial oxidative stress developed in MASLD is appreciated. A comparison with a non-pathological situation of mitochondrial structure, mitophagy, and fatty acid oxidation would be of great interest. Furthermore, I strongly believe that it is really important to improve the image quality of the figures since after zooming in the smaller texts are unreadable. On the other hand, they should mention that there are some minor text errors in the figures. It is important to revise the nomenclature. Here are some examples:

- Mitochondria DNA damage instead of "mitohondria DNA damage" (Figure 1).

- nuclear instead of "necular" (Figure 3).

- FFA instead of "FAA" (Figure 3).

- Peroxisome instead of "Perioxisome" (Figure 3).

- High sugar level instead of "High sugar level" (Figure 3).

Finally, all figures appear in duplicate in the main text.

3.     I suggest that the authors should consider including a summary table or figure outlining the main therapeutic strategies to be addressed in the coming years in the context of the association between mitochondrial oxidative stress and MASLD.

Additionally, the some points to be answered include:

1.     In line 80, I find it unusual to abbreviate free fatty acids as FAA instead of FFA. Check this nomenclature.

2.     As the authors have emphasized, currently no drugs have been approved by the major agencies for the treatment of MASLD. Since mitochondrial dysfunction seems to play an important role in the pathogenesis of the disease, I agree with the authors that targeting mitochondria and modulating mitochondrial function could be a promising field for the treatment of this metabolic disorder. However, some alternatives to consider have been missed by the authors.

A)   On the one hand, semaglutide is showing encouraging results in the control of hepatic steatosis in patients with NASH. In addition, it would be interesting to mention that in Parkinson's patients, semaglutide has been able to protect by reducing oxidative stress and mitochondrial dysfunction by increasing autophagic flux (Liu DX et al. Parkinsons disease 2022). So these types of drugs could be interesting strategies to approach.

B)    Another alternative to consider are strategies targeting the SIRT1/SIRT3 pathway, such as acylated glucagon-like peptide-1 (GLP-1) agonists. These drugs have been shown to ameliorate NAFLD in HFD mice by improving mitochondrial architecture, reducing ROS generation, and inducing autophagy (Tong W. et al. Hepatol. Res. 2016).

C) Third, the authors have mentioned in the main text some relevant data obtained with the mitochondrial antioxidant ubiquinone MitoQ. It would be interesting to mention this option.

D) Interestingly, administration of healthy mitochondria has been shown to have an effect in reducing steatosis and oxidative stress in animals with NAFLD (Fu A. et al. Front. Pharmacol. 2017).

E) Finally, microbiota-based therapies using the liver-gut axis as a target may be of vital importance in these types of metabolic disorders.

3.     Reading this interesting literature review I wonder if there are any papers describing structural differences or differences in oxidative stress resulting from mitochondrial dysfunction between patients with NAFLD or NASH? If not, do the authors believe that there could be differences and would it be important to analyze them to understand disease progression and the involvement of mitochondrial dysfunction? On the other hand, taking into account the recent change of nomenclature of NAFLD by the multi-society Delphi consensus statement, do the authors believe that there could be differences between steatotic liver disease (SLD), metabolic dysfunction-associated steatotic liver disease (MASLD) and MASLD associated with excessive alcohol consumption (MetALD)?

Author Response

In recent decades, MASLD has emerged as one of the most prevalent liver diseases worldwide, in fact it has become a major global health care concern. It is due to the fact that the mechanisms involved in the development of the disease are unknown as well as the lack of approved therapeutic options for the treatment of MASLD. The aim of this literature review is to highlight the role of mitochondrial dysfunction in the development of this disease, in addition to describing possible therapeutic strategies to be addressed in the coming years using mitochondrial dysfunction and oxidative stress as targets. I appreciate very much that the authors have detailed the major structural as well as metabolic modifications associated with mitochondrial dysfunction that are implicated in the development of MASLD. However, I think it is necessary to consider including some issues in the manuscript to achieve an updated and innovative review.

Q1.     I strongly believe that the review requires a section about the interaction between mitochondrial dysfunction and the gut-liver axis. Indeed, due to the relevance of mitochondria-microbiota interaction during NASH, the authors should include the strategy as a potentially promising therapeutic approach to be addressed in the coming years (Juarez-Fernandez M. et al. Hepatology 2023).

Answer for Q1:

Thank you very much for your suggestion. The liver-gut axis plays a vital role in the development of MAFLD. We agree your suggestions and have added this part accordingly. Thereby we discuss how intestinal flora affects mitochondrial function and consequently MAFLD progression in the newly added section 2.7 of the article. In the newly added section 3.4 of the article, we add the current treatment approaches for MAFLD based on beneficial bacteria and intestinal microbial metabolites.

Q2.     The authors' effort to detail the structural mechanism of mitochondrial oxidative stress developed in MASLD is appreciated. A comparison with a non-pathological situation of mitochondrial structure, mitophagy, and fatty acid oxidation would be of great interest. Furthermore, I strongly believe that it is really important to improve the image quality of the figures since after zooming in the smaller texts are unreadable. On the other hand, they should mention that there are some minor text errors in the figures. It is important to revise the nomenclature. Here are some examples:

- Mitochondria DNA damage instead of "mitohondria DNA damage" (Figure 1).

- nuclear instead of "necular" (Figure 3).

- FFA instead of "FAA" (Figure 3).

- Peroxisome instead of "Perioxisome" (Figure 3).

- High sugar level instead of "High sugar level" (Figure 3).

Finally, all figures appear in duplicate in the main text.

Answer for Q2:

 Thank you very much for your guidance and suggestions on the article. I apologize for the jargon issue in the article, and I have fixed it accordingly. The resolution of the pictures in the article has also been improved to better display the details in the pictures. In addition, regarding the problem of repeated images in articles, I think it was caused by my misunderstanding of the image insertion requirements in the submission instructions and I have corrected the image insertion method now.

Q3.     I suggest that the authors should consider including a summary table or figure outlining the main therapeutic strategies to be addressed in the coming years in the context of the association between mitochondrial oxidative stress and MASLD.

Answer for Q3:

Thank you very much for your advices. A summary table of the main treatment strategies is more conducive to reading and comparing the effects of different treatment methods. Therefore, we have added a table (Table 2) in the third part of the article to present the treatment experiments. We hope that the association between mitochondrial oxidative stress and MASLD can be shown more clearly in this table.

Additionally, the some points to be answered include:

  1. In line 80, I find it unusual to abbreviate free fatty acids as FAA instead of FFA. Check this nomenclature.

Answer for 1:

Thank you very much for your question. We have corrected this mistake. And we have checked the expression of other statements in the article to try to avoid such problems.

  1. As the authors have emphasized, currently no drugs have been approved by the major agencies for the treatment of MASLD. Since mitochondrial dysfunction seems to play an important role in the pathogenesis of the disease, I agree with the authors that targeting mitochondria and modulating mitochondrial function could be a promising field for the treatment of this metabolic disorder. However, some alternatives to consider have been missed by the authorsA)On the one hand, semaglutide is showing encouraging results in the control of hepatic steatosis in patients with NASH. In addition, it would be interesting to mention that in Parkinson's patients, semaglutide has been able to protect by reducing oxidative stress and mitochondrial dysfunction by increasing autophagic flux (Liu DX et al. Parkinsons disease 2022). So these types of drugs could be interesting strategies to approach.

Answer for QA:

In the newly added section 3.5 of this review, we respectively supplement the summary about the therapeutic effect of Semaglutide, Liraglutide and other some glucagon-like peptide-1 (GLP) analog for MAFLD.

Semaglutide is a glucagon-like peptide-1 (GLP) analog that can increase lipid metabolism and glucose metabolism in hepatocytes by binding with GLP-1 receptor (GLP-1R), ultimately reducing lipid accumulation and oxidative stress. In addition, by searching the literature, we found that there are currently many other GLP-1 analog drugs used to intervene in the development of MAFLD, such as Liraglutide. Liraglutide inhibits the development of MAFLD by increasing the expression of mitochondrial structural proteins (Drp1, OPA1, and UCP2) and autophagy proteins (Beclin1, LC3, and Sirtuin1). These drugs are currently in clinical phase III experimental testing and are potential targets for the future treatment of MAFLD.  (Liu DX et al. Parkinsons disease 2022; Knudsen LB, et al. Front Endocrinol (Lausanne). 2019; Luna-Marco C et al Redox Biol. 2023;)

B)Another alternative to consider are strategies targeting the SIRT1/SIRT3 pathway, such as acylated glucagon-like peptide-1 (GLP-1) agonists. These drugs have been shown to ameliorate NAFLD in HFD mice by improving mitochondrial architecture, reducing ROS generation, and inducing autophagy (Tong W. et al. Hepatol. Res. 2016).

Answer for QB:

In the newly added section 3.5 of this review, we respectively summarized that the therapeutic effect of S glucagon-like peptide-1 (GLP) analog and other small molecule drugs for MAFLD.

GLP-1 is an incretin hormone that regulates blood sugar and weight homeostasis. Clinical diabetic patients with low expression of GLP-1 receptor (GLP-1R) are known to have mt-ROS accumulation, superoxide formation, and membrane potential loss. In recent years, many GLP-1 analogues have successfully undergone clinical phase II/III validation, such as Liraglutide and Semaglutide. In addition, there are currently some other small molecule drugs that can delay the development of MASH, including FXR agonist, FGF21, PPAR agonist, etc. For example, Efruxifermin (an FGF21 analog) can significantly reduce liver fat accumulation and fibrosis. In addition, the combination of multiple drugs may be more helpful in relieving the diseases than single drug. For example, the combined use of FXR agonist (Cilofexor), ACC inhibitor (Firsocostat), and Semaglutide in MAFLD patients was compared with Semaglutide alone, and the combined use of the drugs improved the two indicators of transaminase and fat content more significantly. (Luna-Marco C et al Redox Biol. 2023; Yang YY et al. Acta Pharmacol Sin. 2022; Tong W. et al. Hepatol. Res. 2016).

C)Third, the authors have mentioned in the main text some relevant data obtained with the mitochondrial antioxidant ubiquinone MitoQ. It would be interesting to mention this option.

Answer for QC:

In the newly added section 3.1.6 of this review, we respectively supplement the functional regulation of Coenzyme Q for MAFLD.

Coenzyme Q (CoQ) is a self-synthesized fat-soluble bioactive quinone, located on the inner membrane of mitochondria and participates in mitochondrial ETM. Clinical data have shown that CoQ10 supplements can alleviate mitochondrial dysfunction induced by MAFLD. Based on your question, we have continued to add information on the therapeutic significance of CoQ in MAFLD. Alhusaini, A. M. et al found that the use of liposome-encapsulated coenzyme Q can significantly reduce liver damage and fiber damage caused by propionic acid though inhibiting cytochrome C, mitochondrial fragmentation to alleviate MAFLD. Tiefenbach, J. et al proposed that coenzyme Q and its analogue-Idebenone can act as PPARα/γ agonists, reduce liver triglyceride and cholesterol levels, and reduce the development of steatosis and MAFLD. Sumbalova, Z. et al. used hydrogen-rich water (HRW) to treat NAFLD clinical patients and mouse models for restoring normal coenzyme Q expression levels, and found that HRW has the potential to regulate mitochondrial oxidation function, and realize the potential of NAFLD treatment. These data reflect coenzyme Q is a potential target to control the progression about oxidative dysfunction of mitochondria in MAFLD.   (Alhusaini, A.M., et al. Int J Mol Sci, 2023; Tiefenbach, J., et al. Dis Model Mech, 2018; Sumbalova, Z., et al. Int J Mol Sci, 2023)

D)Interestingly, administration of healthy mitochondria has been shown to have an effect in reducing steatosis and oxidative stress in animals with NAFLD (Fu A. et al. Front. Pharmacol. 2017).

Answer for QD:

Based on your suggestions, we have understood and summarized the healthy mitochondrial reinfusion treatment in the sections 4.0 of this review. Fu, A. et al. used tail vein injection of healthy mitochondria into MAFLD mice to compensate for the mitochondrial physiological activity of missing functions to alleviate MAFLD. This approach effectively reduced blood lipid content, and significantly restored mitochondrial antioxidant formation. At present, there is still a lack of other animal models or clinical cases to further determine the therapeutic effect of mitochondrial reinfusion, while it is undeniable that it will be a new potential strategy for MAFLD.

E)Finally, microbiota-based therapies using the liver-gut axis as a target may be of vital importance in these types of metabolic disorders.

Answer for QE:

Thank you very much for your suggestions. Mitochondrial function in the liver-gut axis is an important factor affecting the development of MAFLD. In the newly added sections 2.7 and 3.4 of this review, we respectively summarized the functional regulation of mitochondria by microbial flora and their metabolites in MAFLD. And we briefly summarize the treatment of MAFLD by interfering with intestinal microorganisms as following and focusing on beneficial bacteria and intestinal flora metabolites.

Morrison, M. C. et al used heated- Myxiniphila to treat obese mice induced by HFD diet and found that it could significantly decrease intestinal permeability and adipocyte hypertrophy. Whlie Rao, Y. et al treated MAFLD mice with untreated-Proteobacillus myxophila and found that mt- DNA was decreased, and oxidative metabolism markers such as PGC-1É‘ and CPT-1β increased in the hepatocytes. Besides Myxophila up-regulated the expression of mitochondrial complex and activates AMPK activity to sustain lipid oxidation reaction.   (Morrison, M.C., et al. Int J Mol Sci, 2022; Rao, Y., et al. Gut Microbes, 2021)

For Intestinal metabolites such as bile acids (BAs) and short-chain fatty acids (SCFAs), BAs activates FXR and G protein-coupled receptor superfamily (TGR5) to reduce fatty acid synthesis and the activity of mitochondrial complexes IV caused by MAFLD. And Zhao, T.et al found that Butyrate supplementation (a kind of SCFAs) could inhibit the activity of NADH oxidase in the ETC of MAFLD patients and maintain the mitochondrial membrane potential to delay the development of MAFLD.   (Wang, X.X., et al. J Biol Chem, 2022; Morrison, D.J. Gut Microbes, 2016; Zhao, T., et al. Oxid Med Cell Longev, 2020;)

In conclusion, the important regulatory significance of intestinal flora and its metabolites in MAFLD has gradually been valued by researchers, and is expected to become a potential treatment mediated by controlling lifestyle and dietary habits.

  1. Reading this interesting literature review I wonder if there are any papers describing structural differences or differences in oxidative stress resulting from mitochondrial dysfunction between patients with NAFLD or NASH? If not, do the authors believe that there could be differences and would it be important to analyze them to understand disease progression and the involvement of mitochondrial dysfunction? On the other hand, taking into account the recent change of nomenclature of NAFLD by the multi-society Delphi consensus statement, do the authors believe that there could be differences between steatotic liver disease (SLD), metabolic dysfunction-associated steatotic liver disease (MASLD) and MASLD associated with excessive alcohol consumption (MetALD)?

Answer for 3:

Thank you for your questions. First, I believe that mitochondrial dysfunction or oxidative stress has an important impact on the development of MAFLD. The morphology of mitochondria mainly includes characteristics such as size, membrane structure, and permeability.

The specific differences in mitochondrial structure and function in clinical patients with MAFLD. Compared with the typical symptoms of MAFLD, the clinical symptoms of MAFLD are, in addition to ≥5% hepatic steatosis, inflammation accompanied by hepatocytes damage, such as ballooning, inflammatory reaction, fibrosis and other pathological signs. Liver biopsy results of patients with MAFLD usually show that the size of hepatocytes is enlarged to a round shape. Under the electron microscope, loose cytoplasmic matrix, swollen mitochondria, expanded endoplasmic reticulum, broken filaments, or vacuoles, cell membrane thickening and densely stained nuclei can be seen. While the pathological results of MAFLD show that the above structural changes are lighter than those of MAFLD. In addition, the occurrence of MAFLD is related to steatosis, and accumulation of lipid droplets is one of the causes of steatosis. Due to mitochondrial damage, the β-oxidation of fatty acids is affected, resulting in the formation of tiny lipid droplets that accumulates in the cytoplasm. And it has been found in muscle cells that lipid droplets can affect the lipid metabolism pathway between the endoplasmic reticulum and mitochondria. Combining the above research data, we believe that mitochondrial structure and functional disorders are important to the development of MAFLD and are also a potential therapeutic target.   (Bril F, et al. Hepatology. 2017; Bedossa P, et al. Hepatology. 2012; Brunt, E. M.et al. Hepatology.2021; Wozny, M. R et al. Nature 618; Ouyang Q, et al Dev Cell. 2023)

In addition, our team's preliminary research results show that mitochondria in the livers of MAFLD mice are significantly smaller, shorter, and more fragmented. In activated HSCs, the protein expression of mitochondrial fission-related proteins Fis1 and Drp1 was increased significantly, and maximal respiration and spare capacity was increased. Then, when MAFLD mice and activated HSCs were treated with mitochondrial fission inhibitor (Mdivi-1), mitochondria-targeted antioxidant mitoQ, and cellular ROS scavenger Tempol respectively, the degree of liver fibrosis in mice and activated HSCs was significantly alleviated. Based on the above results, we believe that mitochondrial structural damage and oxidative stress state have an important impact on the development of MAFLD. (Zhou, Y., et al. Cell Death Dis, 2022.)

Finally, about the differences between mitochondrial morphological changes and dysfunction in the steatotic liver disease (SLD), metabolic dysfunction-associated steatotic liver disease (MASLD) and MASLD associated with excessive alcohol consumption (MetALD) that you mentioned, we are also very concerned about the differences. Therefore, we newly add a table in the article to list these simple differences from the above aspects of mitochondrial membrane structure, energy production, mt-ROS and mitophagy to supplement this. (Table 1 of this review).

A brief summary is as following. In SLD, the mitochondrial electron transport chain is damaged, such as the activity of mitochondrial complex III is reduced, ATP synthesis is inhibited and mt-DNA content increases. In MAFLD, the mitochondrial membrane permeability increases, the membrane potential decreases and giant mitochondria appear, oxidative phosphorylation and fatty acid oxidation are inhibited, mt-DNA fragmentation and mutations increase. In MeSLD, mitochondrial morphology swells, the inner mitochondrial membrane is damaged, mitochondrial DNA fragmentation increases, autophagy flux increases, and activity Oxygen accumulation, etc.

Reviewer 2 Report

Comments and Suggestions for Authors

This is review paper focused on only one, but may be most important, aspect of NAFLD pathogenesis.  This review shows molecular mechanisms of mitochondrial disruption and presents available preparations, mostly vitamins, that may prevent injury to mitochondria or at least improve its function. 

All 3 figures are shown twice, once with short title and second time with extended description. 

Line 35, chronic hepatitis C is as important as HBV as the cause of HCC

Lines 43 to 48 – insulin resistance and oxidative stress have been mentioned twice, first as hallmarks of NAFLD and again as additional factors. 

Line 66 – “NASH” instead of “NSAH”

Line 660 – “mitochondria” instead of “mitochondrial” 

Discussion as the last chapter of review paper is an odd choice. I think that this subtirle should be changed. Moreover, it is believed that NAFLD pathogenesis is heterogenous with contribution of genetic and environmental factors. The authors should take into account this aspect and  notice hat the role of mitochondrial dysfunction may vary in different patients. 

Clinicians would be interested if there are plasma biomarkers of  mitochondrial dysfunction in NAFLD and what is the differences between lethal mitochondrial injury known to be produced by some withdrawn drugs  and chronic subclinical mitochondrial dysfunction.

The authors should consider change of terminology for NAFLD at to MAFLD or MASLD.   

Author Response

Q1: This is review paper focused on only one, but may be most important, aspect of NAFLD pathogenesis. This review shows molecular mechanisms of mitochondrial disruption and presents available preparations, mostly vitamins, that may prevent injury to mitochondria or at least improve its function.

 All 3 figures are shown twice, once with short title and second time with extended description.

Line 35, chronic hepatitis C is as important as HBV as the cause of HCC

Lines 43 to 48 – insulin resistance and oxidative stress have been mentioned twice, first as hallmarks of NAFLD and again as additional factors.

Line 66 – “NASH” instead of “NSAH”

Line 660 – “mitochondria” instead of “mitochondrial”

Answer for Q1:

Thank you very much for your professional guidance and suggestions for this review. I am very sorry for the technical terminology and incomplete concepts in the article, and now we have searched and corrected them in the section 1 of this review. In addition, regarding the problem of repeated images in the articles, I think the main reason is caused by my misunderstanding of the image insertion requirements in the submission instructions. And then the method of image insertion has been corrected now.

Q2:Discussion as the last chapter of review paper is an odd choice. I think that this subtirle should be changed. Moreover, it is believed that NAFLD pathogenesis is heterogenous with contribution of genetic and environmental factors. The authors should take into account this aspect and notice that the role of mitochondrial dysfunction may vary in different patients.

Answer for Q2:

(1) Thank you very much for your suggestions. It is not suitable to use “discussion” as the final part and analysis in this review. Therefore, we have changed the name of “discussion” to “summary” to better improve the structure of the review.

(2) According to the view you mentioned that "NAFLD pathogenesis is heterogenous with contribution of genetic and environmental factors", we strongly agree with it, and there is no clear comparison and description in the previous article. The pathogenesis of NAFLD or NASH is highly heterogeneous, and often has different reasons in different age structures. For example, for children and adolescents, fatty liver disease mainly induced by genetics, malnutrition, and the environment eventually turns into MAFLD. While in adults, there are many causes, including non-excessive alcohol, genetics, diabetes, etc. Besides these triggers can also result in mitochondrial dysfunction. Comprehensive clinical data analysis found that the number of mitochondria in the liver of patients with MAFLD disease is reduced, and the structure is more obviously damaged than that of MAFLD, which in turn aggravates the process of liver cell necrosis and fibrosis.

Thereby, we have re-edited this part and added it to Part 4 of the review, comparing the causes of disease in adults and children, and tabulating changes about mitochondrial function and structure in different types of fatty liver disease in the newly added Table 1 of this review.

Q3:Clinicians would be interested if there are plasma biomarkers of  mitochondrial dysfunction in NAFLD and what is the differences between lethal mitochondrial injury known to be produced by some withdrawn drugs  and chronic subclinical mitochondrial dysfunction.

Answer for Q3:

(1) Some studies have shown that lactic acid, pyruvate, alanine, aspartic acid, glutamic acid, glutamine, glutathione, malondialdehyde and other substances in serum may be related to liver function and mitochondrial dysfunction. Mitochondrial dysfunction can also be detected by the size, density, and composition of extracellular vesicles in the peripheral blood of MAFLD patients. In addition, studies also discovered three protein biomarkers for mitochondrial disease detection, including mitochondrial carbamoyl phosphate synthase 1 (CPS-1), Ornithine transcarbamoylase (OTC), and Fibroblast growth factor-21 (FGF-21). However, these indicators cannot directly detect and explain the functional changes of mitochondria in the liver, and still need to be futher used with other liver functional indicators to distinguish.

(Thietart S. J Hepatol. 2020; Peñas A et al. Int J Mol Sci. 2021; Lee, A. H.et al. Front Endocrinol (Lausanne). 2022; Ajaz S.et al. Mitochondrion. 2021)

(2) Differences in mitochondrial structure and function exist in patients with chronic fatty liver disease and drug-induced liver injury. In this regard, we selected several main functions for comparison in the Table 1 of this review.

The outer mitochondrial membrane disappears, the mitochondrial membrane potential decreases, ATP is decreased, the copy number of mitochondrial DNA is reduced, and mitochondrial selective autophagy occurs in patients with drug-induced liver injury (DILI). In patients with simple fatty liver disease (SLD), the mitochondrial electron transport chain is damaged, ATP synthesis is inhibited, and mt-DNA content is increased. In patients with Metabolic dysfunction-associated SLD(MASLD), mitochondrial membrane permeability is increased, membrane potential is decreased, giant mitochondria appear, and mt-DNA fragments in hepatocytes. In patients with Excessive alcohol and metabolic-associated SLD (MetALD), mitochondrial swells, mitochondrial inner membrane is damaged, and mitochondrial DNA fragments, increasing autophagy and the accumulation of reactive oxygen species.

Q4:The authors should consider change of terminology for NAFLD at to MAFLD or MASLD.   

Answer for Q4:

Thank you very much for your suggestions. In 2020, NAFLD was renamed MAFLD. This reflects that the metabolic disorders have significant impact on the development of fatty liver disease. Then, we have corrected NAFLD as MAFLD in the review.

Reviewer 3 Report

Comments and Suggestions for Authors

The Authors should dedicate a paragraph to the oxidant stress and its relationship to the endotoxin levels in NAFLD

Comments on the Quality of English Language

Moderate editing of English language is required.

Author Response

Q: The Authors should dedicate a paragraph to the oxidant stress and its relationship to the endotoxin levels in NAFLD

 Answer for Q:

Thank you very much for your professional guidance and suggestions for this review. Endotoxin produced by intestinal flora microorganisms is one of the important factors affecting the development of MAFLD. So now we have newly added these contents to this review in the third part of Section 2.7 and revised the sentences in the full text seriously according to your suggestions.

After the intestinal mucosal barrier was damaged, endotoxin could transfer into liver tissue and continuously induce the release of pro-inflammatory cytokines and promote oxidative stress in hepatocyte, promoting the development of MAFLD. According to studies, the relationship between oxidative stress and endotoxin regulation in MAFLD is complex. On the one hand, oxidative stress can increase intestinal permeability to lead to more endotoxin transfer into the blood and liver, finally exacerbating liver inflammation and fibrosis. On the other hand, endotoxin can induce oxidative stress in the liver, further damaging hepatocytes by activating the NLRP3 inflammasome and releasing proinflammatory factors such as IL-1β (Cho, Y.E., et al., Hepatology, 2021; Verma, K., et al., Curr Res Pharmacol Drug Discov, 2022; Seo, H.Y., et al., PLoS One, 2023)

Round 2

Reviewer 1 Report

Comments and Suggestions for Authors

I appreciate the effort made by the authors to make all the modifications and suggestions, as well as the time taken to address all the proposed issues. I strongly believe that the new sections "Gut microbiota in the liver-gut axis influence mitochondrial function" and "Targeted microbiota therapies targeting the liver-gut axis" provide an innovative perspective with respect to other reviews that can be found in the literature.

Author Response

We would like to thank you for your professional reviewwork,constructive comments,and valuable suggestions on our manuscript.